# The state of memory-matched distractor in working memory influence the visual attention

Quanshan Long[1]◉, Ting Luo[2]◉, Sheng Zhang[1], Yuanling Jiang[3], Na Hu[4], Yan Gu[1], Peng Xu[3], Antao Chen[1]*

**1** Key Laboratory of Cognition and Personality of Ministry of Education, Faculty of Psychology, Southwest University, Chongqing, China, **2** Department of Psychology, Jiangsu Normal University, Xuzhou, Jiangsu Province, China, **3** The Key Laboratory for NeuroInformation of Ministry of Education, School of Life Science and Technology, University of Electronic Science and Technology of China, Chengdu, China, **4** Department of Preschool & Special Education, Kunming University, Kunming, China

◉ These authors contributed equally to this work.
* xscat@swu.edu.cn

**Data Availability Statement:** Data are available in Open Science Framework (DOI 10.17605/OSF.IO/U2PBK).

**Funding:** A.T. C. received the grants from the National Natural Science Foundation of China (61431013, 31771254) and the Fundamental

## Abstract

Information in working memory (WM) can guide visual attention towards matched features. While recent work has suggested that cognitive control can act upon WM guidance of visual attention, little is known about how the state of memorized items retaining in WM contribute to its influence over attention. Here, we disentangle the role of inhibition and maintenance on WM-guided attention with a novel delayed match-to-sample dual-task. The results showed that active inhibition facilitated searching by diminishing sensory processing and deterring attentional guidance, indexed by an attenuated P1 amplitude and unaffected N2pc amplitude, respectively. By contrast, active maintenance impaired searching by attentional guidance while sensory processing remained unimpaired, indexed by an enhanced N2pc amplitude and unchanged P1 amplitude, respectively. Furthermore, multivariate pattern analyses could sucessfully decode maintenance and inhibition, suggesting that two states differed in modulating visual attention. We propose that remembered contents may play an anchoring role for attentional guidance, and the state of those contents retaining in WM may directly influence the shifting of attention. The maintenance could guide attention by accessing input information, while the inhibition could deter the shifting of attention by suppressing sensory processing. These findings provide a possible reinterpretation of the influence of WM on attention.

## Introduction

In recent years, the interaction between working memory (WM) and attention has become a topic of considerable interest [1–3]. Some researchers have indicated that the maintained contents automatically guide visual attention [4–6], while some other researchers have found that presenting a memory-matched distractor did not capture individuals' attention [7, 8]. These finding imply that there are different WM states which may have distinct impacts on visual attention. Olivers et al. (2011) proposed that WM representations may be in one of two different states: active and accessory. The active items are within the focus of executive processes

Research Funds for the Central Universities of
China (SWU1609106; SWU1709107;
SWU1809361). The funders had no role in study
design, data collection and analysis, decision to
publish, or preparation of the manuscript.

**Competing interests:** The authors have declared
that no competing interests exist.

and become an attentional template for guiding attention during visual tasks. By contrast, the
accessory items are peripheral to the current mental manipulation and have relatively little
influence on selective attention.

An intriguing question is whether the state of memorized items retaining in WM (e.g., inhibition and maintenance) contribute to this attentional guidance when the memory-matched items
are presented as distractor in the search task. As explained earlier, the different WM representations may have different degrees of maintenance, implying that the state of memorized items
retaining in WM play a critical role in the guidance effect. Remembered content and maintenance
state (indicating the to-be-maintained content that is task-relevant) influence selective attention
in the same direction (guiding attention to memory-matched information). Notably, the other
WM state is inhibition, which can lead to rejecting or suppressing specific (e.g., irrelevant) information in WM [9]. In this case, participants may set up a "negative template" and avoid attending
to these memory-matched contents, which may in turn improve distractor supression and therefore benefit target detection [7, 10, 11]. Thus, the inhibition state and remembered content may
have opposite influences on selective attention: the former may force attention away from an
memory-matched content, while the latter will direct attention to that item.

Olivers et al. (2006) reported that inhibited WM content matching a distractor resulted in
a trend to facilitate visual attention. In their dual task, two consecutively presented colors were
memorized; then, a color was cued to become relevant (to-be-maintained content) while another
became irrelevant (to-be-inhibited content) for a subsequent memory test. Thus, it is possible
that WM formed two separate memory templates based on the two colors. Because only one
object can be actively maintained in WM at a time [3], participants might focus on the template
of the to-be-maintained content. In this case, the operation on the template of the to-be-inhibited
content should be passive and weak, which may account for the non-significant facilitation.

In the present study, we planned to use a revised dual task to investigate the influences of
the state (especially, inhibition) of memorized items retaining in WM on visual attention. Specifically, two colors were first memorized simultaneously, and thereafter respectively cued as
to-be-maintained content (cued condition) and to-be-inhibited content (uncued condition) in
WM (see Fig 1), ensuring that they were remembered in the same WM template. In this case,

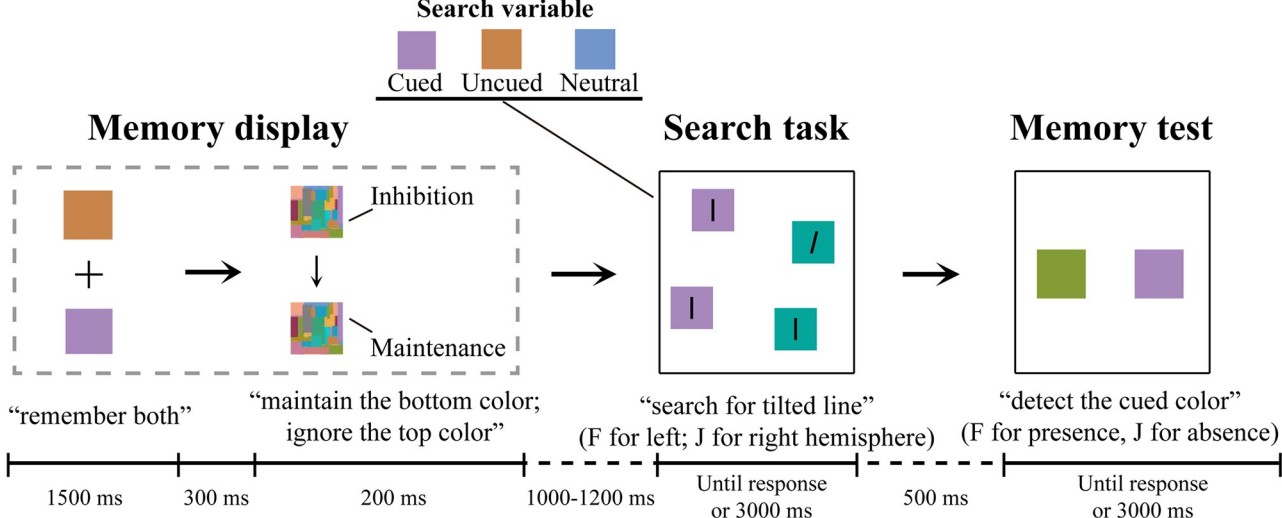

**Fig 1. Schematic illustration of the trial sequence, stimulus displays, and response requirements in the present study.** The Search variable (cued,
uncued, and neutral colors) was presented contralateral to the search target.

participants were supposed to actively conduct both maintenance and inhibition on this template, which may benefit from top-down modulation of sensory processing [12]. Accordingly, the influence of inhibition on selective attention could be enhanced and revealed.

We recorded behavioral and EEG data of visual attention modulated by WM contents with different states, and focused on two ERP components: P1 and N2pc. P1 is a positive deflection peaking at about 100 ms, indexing selective attention at the sensory gating stage [13, 14]. N2pc is a negative-going voltage deflection that is typically observed around 200 ms after the onset of a visual search stimulus, which is sensitive to the shifts of selective attention [15–17]. Given that multivariate pattern analyses (MVPA) [18] can combine all characteristics of the data to help us understand the difference between inhibition and maintenance from the multivariate level, we used MVPA to maximally discriminate two WM states in modulating attention. We expected that they would result in distinct modulation patterns. The findings of this study not only highlight the role of WM state in the guidance of WM on visual attention, but may also open a new avenue to illuminate the interaction between WM and attention.

## Methods

### Participants

A priori statistical power analysis (power > 0.95) was performed based on assuming a small-to-medium effect size of 0.25 [19]. The projected minimum sample size needed was 14 participants. Considering the sample sizes of published studies, twenty-four (11 females, age from 18 to 23 years) right-handed college students were recruited and monetarily rewarded for participation. Data from two participants were excluded because of excessive errors (exceeding three standard deviations) in search performance. All participants reported normal or corrected-to-normal visual acuity and normal color vision. The written informed consent was obtained from each participant. This study was approved by the Review Board of Southwest University (China) for Human Participant Research. All procedures performed in studies involving human participants were in accordance with the ethical standards of the institutional and/or national research committee and with the 1964 Helsinki declaration and its later amendments or comparable ethical standards.

### Design

The state of WM manifesting in the search task were manipulated. Specifically, the contents remembered in WM appeared in the search task in half of the trials, among which the cued color (asking for maintenance) and the uncued color (asking for inhibition) were presented in the opposite visual field to the search target with equal probability. In the other half of the trials, colors other than the cued or the uncued color were displayed in the search task, which served as the neutral condition. Therefore, three levels of the Search variable (cued, uncued, and neutral conditions) were established in a 1:1:2 proportion, to avoid any bias to the memory contents. In the memory test, the cued and the uncued color appeared equally often and independently of each other, thus generating four combinations according to whether the cued and uncued colors were presented.

The participants in the practice session completed 24 trials with feedback on the memory test, which were excluded from the data analysis. Then, the experiment with 24 blocks of 24 trials was performed. Trials were pseudo-randomly assigned. Each block took 2–3 minutes, and participants had self-determined rest breaks.

## Apparatus, stimuli, and procedure

E-Prime 1.1 (Psychology Software Tools Inc. Pittsburg, USA) was used to control the stimuli and collect responses. The stimuli were displayed on a 17-inch monitor with a resolution of 1,024 × 768 pixels and a refresh rate of 85 Hz. The distance between the monitor and the participants was approximately 70 cm. Stimuli were presented on a gray background (RGB values: 128, 128, 128). The color patch was a 1.6° × 1.6° visual angle, in the middle of which was a black line (0.8° × 0.1° visual angle) vertically presented (distractor) or tilted 15° (target). Ten colors were chosen from Munsell's color system [20] with five principal hues (red, yellow, green, blue, and purple). Table 1 listed the hue, brightness, and chroma of these colors. The hue and value were nearly constant. The chroma of each color varied between 8 and 12.

The procedure scheme was displayed in Fig 1. Each trial began with a black cross presented for 500 ms. After that, the memory display with two different colors was presented for 1500 ms, and participants were required to remember both colors. The color patches were presented 1.5° up and down from the center fixation. The color patches were replaced by Mondrian patches for 500 ms, constituting the mask display. In the last 200 ms of the mask display, an upward or a downward arrow appeared randomly with equal probability on the fixation. The arrow pointed to either of the Mondrian patches where the masked color was to be tested in the later memory test, introducing the maintenance state on the cued color and the inhibition state on the uncued color. An inter-stimulus interval randomized between 1000 ms and 1200 ms was added before the search task. The search task contained four color patches separately in four quadrants on an imaginary clock face with a radius of 6.5°. Two different colors were chosen and assigned to the left and the right visual field, respectively, in which the color in the same visual field was identical. The cued color was located equally in the left and the right visual fields, as was the uncued color. The search target was presented in the four quadrants evenly. Participants discriminated the location of targets where the tilted line appeared by pressing keyboard ("F" for left side of the screen, "J" for right side of the screen). Notably, the search targets were always presented contralateral to the cued or uncued colors. After an interval of 500 ms, two different color patches were presented horizontally on the memory test, for which the participants were required to report whether the cued color was presented: "F"/"J" for the presence; "J"/"F" for its absence. It must be

**Table 1. Details of the colors used in the experiment.**

|  | Number | Munsell (as chosen) | | | CIE (as measured) | | |
|---|---|---|---|---|---|---|---|
|  |  | Hue | Value | Chroma | x | y | Y |
| Yellow | 1 | 8Y | 6 | 8 | 0.4297 | 0.4738 | 30.05 |
| Yellow-red | 2 | 8YR | 6 | 11 | 0.5009 | 0.4316 | 30.05 |
| Red | 3 | 8R | 6 | 12 | 0.4999 | 0.3476 | 30.05 |
| Red-purple | 4 | 8RP | 6 | 12 | 0.4172 | 0.2814 | 30.05 |
| Purple | 5 | 8P | 6 | 12 | 0.3136 | 0.2243 | 30.05 |
| Purple-blue | 6 | 8PB | 6 | 12 | 0.2281 | 0.1980 | 30.05 |
| Blue | 7 | 8B | 6 | 9.5 | 0.1996 | 0.2400 | 30.05 |
| Blue-green | 8 | 8BG | 6.6 | 8.2 | 0.2212 | 0.3125 | 37.52 |
| Green | 9 | 8G | 6.3 | 9 | 0.2438 | 0.3852 | 33.66 |
| Green-yellow | 0 | 8GY | 6 | 12 | 0.3398 | 0.5548 | 30.05 |

*Note*. The Munsell values (red/green/yellow/purple/blue) were converted to red, green, and blue (RGB) values using the Munsell Conversion computer program [21]. The current colors and the original Munsell colors were not an exact match because of limited color space on the computer screens. CIE refers to Commission Internationale de L'Eclairage values.

noted that the button responses were counterbalanced across participants in the memory test. Both the search task and the memory test were presented until a response was obtained or for 3000 ms. The inter-trial interval was 500 ms.

The colors presented in the dual task were controlled in certain respects. In the memory display, colors were chosen from two adjacent hues (red, yellow, green, blue, and purple) to ensure visual similarity, which was supposed to ensure visual representation [5]. In the search task, when the cued or the uncued color appeared as one color, the other color was the most dissimilar one to it. Otherwise, two different colors dissimilar to both the cued or the uncued color were displayed. Additionally, the colors were approximately equated with luminance contrast and color distance to avoid differences in physical attributes.

## EEG recordings and pre-processing

The EEG data were recorded using a Brain Products system (band pass 0.01–100 Hz, sampling rate: 500 Hz, notch off) that was connected to a 64 scalp Ag-AgCl electrodes placed according to the international 10–20 system. All inter-electrode impedances were kept below 5 kΩ with the references on FCz and a ground electrode on AFz. The EOGs were simultaneously recorded from four surface electrodes, which were placed over the upper and lower eyelids, and then by the outer canthus of the left and right eyes to monitor ocular movements and eye blinks.

Offline EEG data were re-referenced to the bilateral mastoid electrodes and filtered with a band-pass filter of 0.05–40 Hz by using EEGLAB [22]. Continuous EEG data were segmented from -200 to 1000 ms relative to all stimulus markers. The baseline was corrected using the pre-stimulus time interval (-200 ms). Independent component analysis (ICA) decomposition was run on this dataset to remove ocular artifacts and additional noise [23]. In order to improve ICA decomposition, an initial rejection of noisy EEG data was performed by using visual inspection. Trials contaminated by gross movements were rejected. Automated detection and removal of artifactual independent components (e.g., blinks, eye-movements) were accomplished using the ADJUST toolbox [24]. ERP amplitudes were averaged across trials for each Search variable.

## Data analyses

For all analyses, Greenhouse-Geisser correction for degrees of freedom was used whenever the assumption of sphericity was violated ($p < .05$). For multiple comparisons, the Holm-Bonferroni correction was used to adjusted $p$ value.

## Behavioral analyses

In the memory test, to exclude the influence of other confounders, only data from the neutral condition in search task were entered in the analysis of accuracy and reaction time (RT). All data were analyzed with a 2 (Presence of cued: Cued_presence, Cued_absence) × 2 (Presence of uncued: Uncued_presence, Uncued_absence) repeated-measures ANOVA.

In the search task, data from trials in which the search task and memory test were correct were entered in the analyses of RT and ERP. Trials with RT exceeding three standard errors for each participant were removed, leaving at least 104 trials for each condition for the analysis of RT. One-way repeated-measures ANOVA was carried out on the Search variable.

## Univariate analyses

The P1 and N2pc components were analyzed in the search task. The P1 component peaked at around 100 ms poststimulus, which was analyzed in 60 to 120 ms interval to examine whether

the states of remembered items retaining in WM influence the early visual procedding [13, 25]. The P1 component was analyzed at the lateral posterior electrode sites (PO7, PO8). A 2 (Lateral: ipsilateral, contralateral) × 3 (Search variable: cued, uncued and neutral) repeated-measures ANOVA was executed on P1 amplitudes.

The N2pc component was sourced at the pooled occipital electrodes (PO7/PO8) and obtained by subtracting the ipsilateral waveforms from the contralateral waveforms of the target, which could explore the attentional guidance for different WM states [15, 17]. Thus, the N2pc amplitude was analyzed in the Search variable with a one-way ANOVA. Additionally, the mean amplitudes averaged across 170–250 ms were compared between the contralateral and ipsilateral electrodes with paired-sample *t* tests for each Search variable, to estimate the guidance effect of the WM.

## Multivariate pattern analyses

The MVPA was performed to maximally discriminate between inhibition and maintenance. A cross-subject approach was used in the present study, in which a classifier was trained on the data of 21 subjects and then applied to the remaining subject by using the leave-one-out cross-validation method. We applied an MVPA combining feature selection techniques with a support-vector machine (SVM) by using the LIBSVM toolbox [26]. The exceptionally high dimensionality of the data was reduced by selecting the most relevant features. Considering the main purpose of the current study, we chose attention-related components (P1, N2pc) to train the SVM classifier, as these components were closely related to visual attention [4, 14].

We employed attention-related features for which we combined data across corresponding electrodes (P1 amplitude: averaged across electrodes PO7 and PO8; N2pc amplitude: contra-lateral—ipsilateral waveforms) and time intervals of two ERP components. Each participant contained two WM states, and the averaged ERP waveforms at given time windows and corre-sponding electrodes were fed into the SVM classifier. To improve the signal-to-noise ratio, the data from a given state were averaged across trials, and decoding was performed on these aver-ages rather than on single-trial data. Thus, two averaged ERP difference waves (maintenance = cued—neutral; inhibition = uncued—neutral) that can eliminate baseline interference served as classification features to decode maintenance and inhibition. To investi-gate whether decoding accuracy would be higher than chance level (50%), a permutation test was performed for the classification of WM state. To this end, each dataset was randomly labeled as maintenance or inhibition. The classifier was trained on these features. The random-ization process was repeated 1000 times, and then the distribution of classifications was com-pared with the classification based on the real data.

Furthermore, we compared the predictive value of the averaged ERP difference wave approach with additional approaches that do not consider the influence of the baseline or both ERP amplitudes. The decoding procedure for additional approaches was the same as the above approach.

# Results

## Behavioral results

**Memory test.** Analyses of the accuracy showed a significant Presence of cued × Presence of uncued interaction (Fig 2a; $F(1,21)$ = 6.53, $p$ = .018, $\eta_p^2$ = 0.24). Follow-up tests demon-strated that participants could obtain a high accuracy rate when the cued color was present in the memory test, regardless of whether the uncued color was present (90.3% vs. 92.7%; $F(1,21)$ = 3.47, $p$ = 0.076). However, when the cued color was absent, participants showed worse per-formance for the presence of the uncued color than its absence (73.7% vs. 81.1%; $F(1,21)$ =

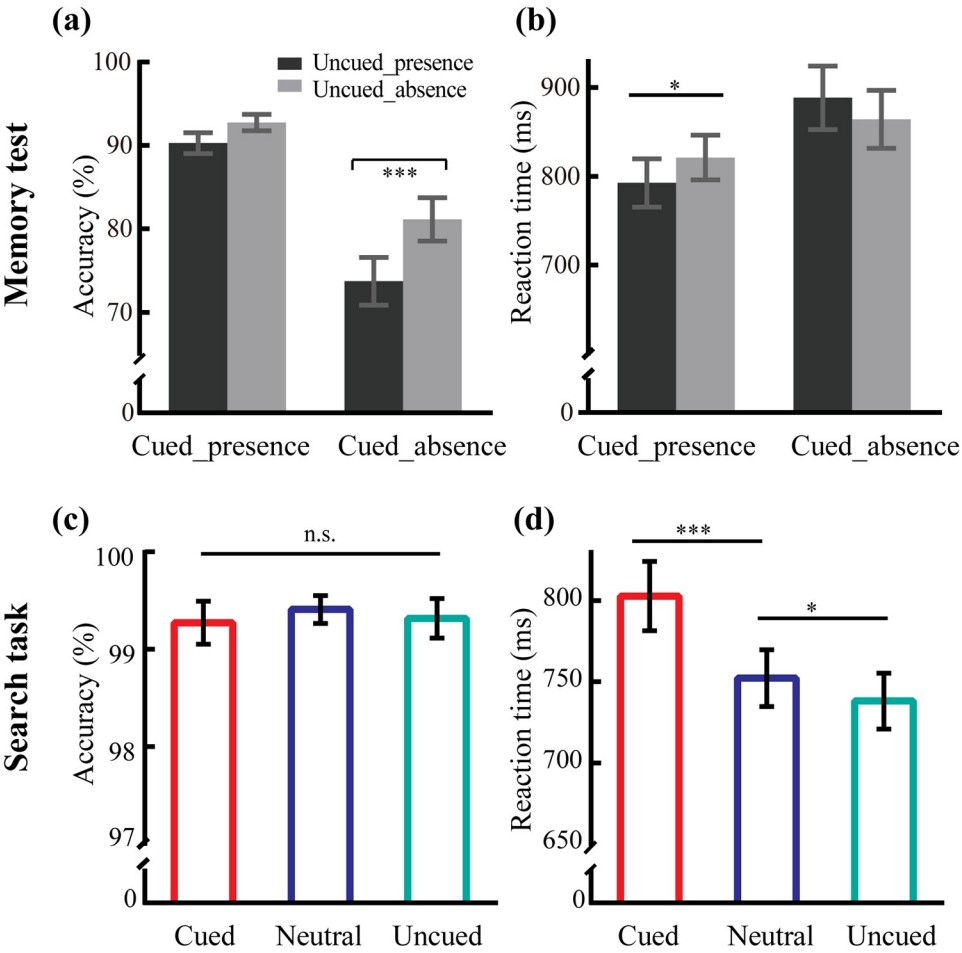

**Fig 2. Results of reaction time and accuracy in the search task and memory test, respectively.** Error bars denote the standard error of the mean (SEM). n.s. = non-significant, * $p < .05$, *** $p < .001$.

30.73, $p < .001$, $\eta_p^2 = 0.59$). Also, the main effect of Presence of cued was significant ($F(1,21) = 24.71$, $p < 0.001$, $\eta_p^2 = 0.54$), with higher accuracy in the presence (91.5%) than in the absence of the cued color (77.4%). And the main effect of Presence of uncued were significant ($F(1,21) = 29.64$, $p < 0.001$, $\eta_p^2 = 0.58$), with lower accuracy in the presence (82.0%) than in the absence (86.9%) of the uncued color. Furthermore, when only the cued or uncued color was presented in the memory test, participants showed better performance for the presence of the cued color relative to the presence of the uncued color ($t(21) = 6.44$, $p < 0.001$).

Only the correct trials were entered in the analysis of RT. There was a significant interaction between Presence of cued and Presence of uncued (Fig 2b; $F_{(1,21)} = 10.73$, $p = .004$, $\eta_p^2 = 0.34$). When the cued color was presented in the memory test, participants tended to respond faster in the presence than in the absence of the uncued color (793 vs. 821 ms; $F_{(1,21)} = 7.75$, $p = .011$, $\eta_p^2 = 0.27$). In contrast, when the cued color was absent, the difference between the presence and the absence of the uncued color was not significant (889 vs 864 ms; $F_{(1,21)} = 2.64$, $p = .119$). The main effect of Presence of cued was significant ($F(1,21) = 10.57$, $p = 0.004$, $\eta_p^2 = 0.34$), with faster response in the presence (807 ms) than in the absence (876 ms) of the cued color. The main effect of Presence of uncued was not significant ($F(1,21) = 0.05$, $p = 0.831$). Additionally, if we merely presented the cued or uncued color in the memory test, participants

showed faster responses in the presence of the cued color than in the presence of the uncued color ($t_{(21)}$ = -2.86, $p$ = 0.009).

**Search task.** Participants obtained high accuracy (over 99%) in all Search variables (Fig 2c; $F_{(2,42)}$ = 0.23, $p$ = .799). The mean RT for three conditions was 764 ms. There was a significant main effect of the Search variable on the measures of RT (Fig 2d; $F_{(1,31)}$ = 33.94, $p$ < .001, $\eta_p^2$ = 0.62). The RT in the cued condition (803 ms) was slower than those in the neutral (752 ms; $p$ < .001) and uncued conditions (738 ms; $p$ < .001). Importantly, the RT in the uncued condition was significantly better than those in the neutral condition ($p$ = .024). These results indicated that the appearance of the cued color opposite to the search target significantly hindered selection of the target, while the presence of the uncued colors significantly facilitated visual search.

## Univariate analyses: P1 and N2pc

The analysis of P1 amplitude showed significant main effects of the Search variable ($F_{(2,42)}$ = 4.37, $p$ = .019, $\eta_p^2$ = 0.172). Pairwise comparisons of the Search variable (see Fig 3) showed smaller P1 amplitudes in the uncued condition (0.466 µV) than in the neutral condition (0.78 µV; $p$ = .028), but no significant difference between the cued (0.77 µV) and neutral conditions ($p$ = .92). The main effect of Lateral was not significant ($F_{(1,21)}$ = 0.18, $p$ = .674). P1 amplitudes induced by three conditions were not lateralized, and equally present at electrodes ipsilateral and contralateral to the target. The interaction between Lateral and the Search variable was not significant ($F_{(2,42)}$ = 0.007, $p$ = .993).

As indicated by Fig 4a, 4b and 4c, the N2pc amplitude was strongly affected by the cued colors presented as distractors, with large amplitudes for contralateral waveforms than ipsilateral waveforms in the cued condition ($t_{(21)}$ = 5.00, $p$ < .001). However, no difference emerged in the uncued ($t_{(21)}$ = 1.13, $p$ = .273) and neutral conditions ($t_{(21)}$ = -0.57, $p$ = .574). These observations were confirmed by the ANOVAs on the mean difference waves in the N2pc time window (170–250 ms). This analysis showed a remarkable main effect of the Search variable (Fig 4d; $F_{(2,42)}$ = 20.08, $p$ < .001, $\eta_p^2$ = 0.49), indicating that the N2pc amplitudes were larger for the cued condition (-0.881 µV) than for the uncued (-0.131 µV; $p$ < .001) and neutral conditions (0.052 µV; $p$ < .001); no difference emerged between the uncued and neutral conditions ($p$ = .17).

## MVPA analyses

MVPA were used to predict the WM states (maintenance or inhibition) based on attention-related ERP responses. First, the SVM classifier was trained on both ERP difference waves, resulting in a maximum accuracy of 68.18%. Permutation tests confirmed that this accuracy was significantly greater than chance (50%; $p$ = 0.028).

Next, we compared our approach with one that did not consider the influence of neutral condition or both attention-related ERP components. We therefore averaged the ERP amplitude for each Search variable instead of calculating the difference wave between the cued/uncued and neutral conditions. This approach resulted in a maximum accuracy of 68.18% ($p$ = 0.032). Then, we conducted a classification based on the single ERP amplitude or single difference wave. Classification based on single P1 or N2pc difference waves showed a maximum accuracy of 61.36% and 70.45%, respectively. Besides, single P1 or N2pc amplitude served as classification features, resulting in a maximum accuracy of 50% and 65.91%, respectively.

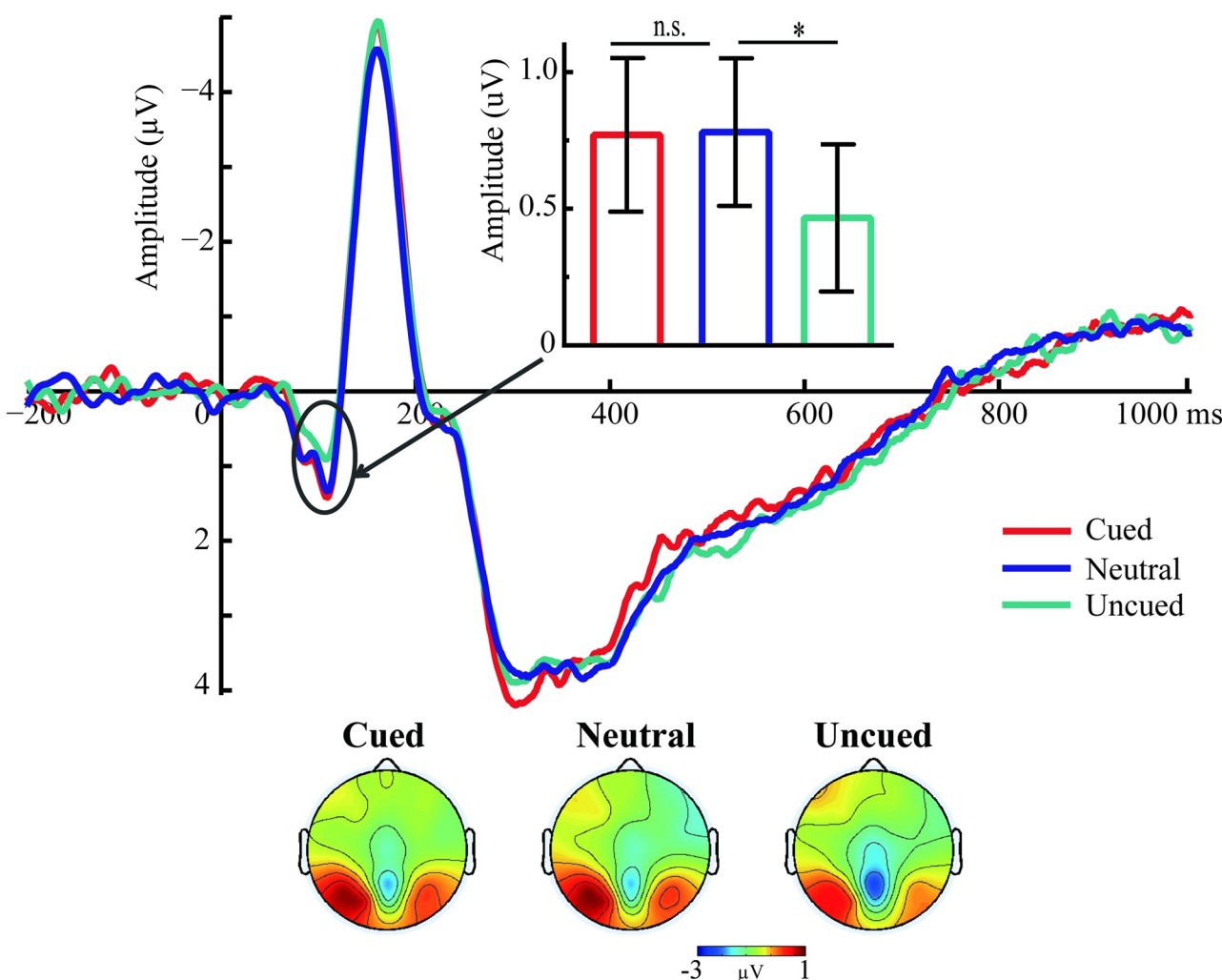

**Fig 3. The P1 amplitudes for the search variable at the posterior electrode sites (averaged across electrodes PO7 and PO8) are presented.** Topography represents the P1 component measured as the mean amplitude between 60 and 120 ms. The left and right sides of the outline head represent left and right hemisphere locations, respectively. n.s. = non-significant, * $p < .05$.

Overall, multivariate assessment of the ERP patterns allowed for significant predictions of the modulating pattern of WM states from brain activity, suggesting inhibition and maintenance are significantly different with each other in modulating visual attention.

## Discussion

We used a new delayed match-to-sample procedure, in which both maintenance and inhibition of WM were actively executed, and recorded EEG data to investigate the influences of the WM states on visual search. The behavioral results showed that the time of visual search significantly increased in the cued condition but significantly speeded up in the uncued condition, compared to the neutral condition. Furthermore, the ERP results indicated that P1 amplitude was significantly attenuated in the uncued condition relative to the neutral condition, but was not different between the cued and neutral conditions. Interestingly, N2pc was significantly enhanced in the cued condition compared with the neutral and uncued conditions, but did not differ between the latter two conditions. Besides, the two WM states could be decoded

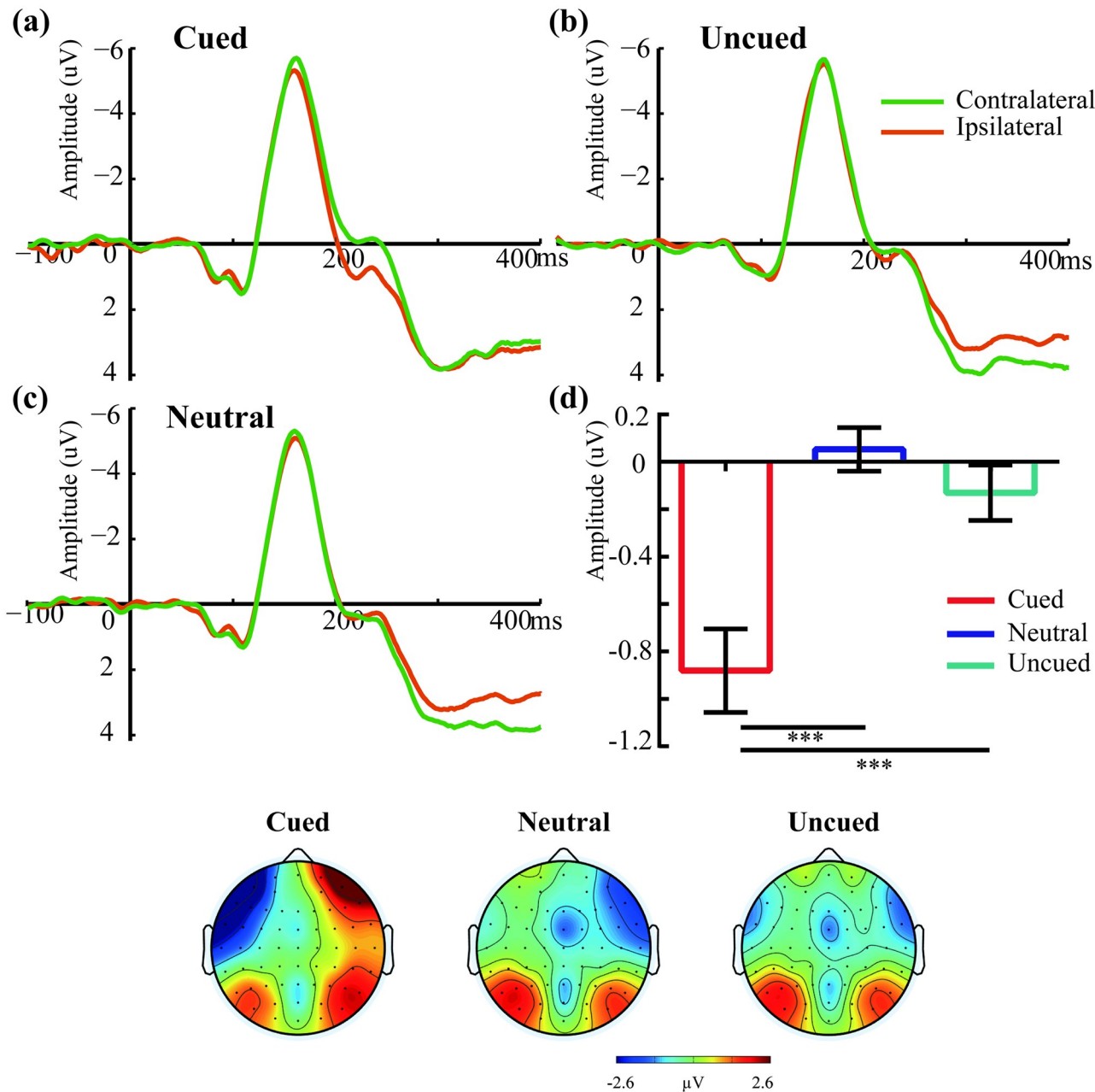

**Fig 4. The N2pc waveforms (a, b, c) for contralateral and ipsilateral electrical activity and mean difference wave (d) are presented.** Topography represents the N2pc component (170–250 ms). The left and right sides of the outline head represent ipsilateral and contralateral electrode sites of the target, respectively. *** $p < .001$.

using the SVM classifier, suggesting they were significantly different in modulating visual attention.

In the uncued condition, one color of the search display matching the uncued color (to-be-inhibited color in WM) was always located on the side contralateral to the target. To maximally reduce interference from the to-be-inhibited color in WM, the uncued color served as a template for rejection (negative template) that biased attention away from the contralateral side during visual search [10, 27–29], which in turn benefited attention to the ipsilateral side.

Attenuated P1 amplitude in the uncued condition relative to the neutral condition suggested that the inhibition of WM diminished sensory processing [30, 31], which may have eliminated the guidance effect in the uncued condition; owing to this, the N2pc amplitude was comparable between the uncued and neutral conditions [15]. It is possible that diminished sensory processing helped participants effectively avoid noticing the contralateral color; in this case, their attention would not be guided to the contralateral side. Moreover, relative to the neutral condition, the uncued condition facilitated visual search behaviorally. Because the interference signal coming from uncued color matched location would obscure the signal coming from the location of search target, and the active suppression of interference signal would lead to improved behavioral accuracy (facilitation effect). In this case, attenuated P1 may reflect the mechanism of active suppression deriving from the inhibition [25].

The facilitation effect demonstrated that the to-be-inhibited color in the uncued condition was still represented in WM during the search task [16, 32], which was also supported by the finding that when the to-be-inhibited color appeared in the subsequent memory test, the test for the to-be-maintained color was significantly affected. Traditionally, content remembered in WM is thought to enable guidance of visual attention; if this is the case, then visual search would be impaired in the uncued condition. The overall pattern of facilitation in the uncued condition suggested that the facilitation effect induced by the inhibition of WM was stronger than the impairment effect induced by the remembered color; although, it is possible that the state of memorized items retaining in WM rather than WM content influenced visual attention. Regardless of which explanation will be verified, the current results clearly showed that WM representation is essential for guidance, in which it may play an anchoring role in attentional guidance. That is, only when one color was remembered in WM could the WM states (inhibition and maintenance) influence visual attention. Moreover, the current results highlighted the necessity of separating the two aspects of one WM representation (the remembered content and the state of those items retaining in WM) when studying how WM guides attention, because the evidence indicated that they make different but complementary contributions to guidance. Besides, although both colors were stored in WM representation, to-be-maintained color had larger bias signal with it under the model of Bundesen's theory of visual attention (TVA) [33], which might generate attentional guidance; but to-be-inhibited color had a smaller attentional weight compared to neutral and to-be-maintained colors, which may facilate the search performance shown in Fig 2d. It is worth noting that this weight difference might be caused by maintenance and inhibition, suggesting the importance of WM states in guiding visual attention.

In accordance with previous studies [1, 4, 5, 8], our behavioral results showed that the to-be-maintained color in the cued condition impaired the visual search. In the present study, the memory-matched color was a distractor that was always presented contralateral to the search target. The to-be-maintained items might be viewed as a template for rejection in the search task [10], while these items should be maintained in the working memory task. In this case, the requirement of working memory and search demand engender different response, evoking a conflict [34–36]. Therefore, this conflict hindered the effect of rejection template, leading to slower response time in the cued condition than in the neutral condition. In our opinion, the maintenance state in the cued condition allowed participants' attention to access input from a color matching the remembered color in WM. Since the color matching, attention would be directed to the contralateral side, indexed by the enhanced N2pc amplitude in the cued condition relative to the neutral condition, which therefore impaired the visual search. In the present study, the cued items induced larger N2pc amplitudes in contralateral side than in ipsilateral side; but no difference was found for the uncued/neutral items between the two sides. Thus, different from the cued condition, the uncued and neutral conditions would not capture

attention at all. Here, the remembered color in WM could still play an anchoring role for the shifting of attention. Although shifting attention to the contralateral side is disruptive for the search task, access from the remembered color to the input color kept by the maintenance state favored the normal sensitivity of sensory processing in the cued condition, indexed by unchanged P1 amplitude between the cued and neutral conditions. Thus, despite the detrimental influence on the visual search, a shift of attention did occur in the cued condition, which also indicated the automaticity feature of guidance induced by WM representation [37]. From the current results, we further propose that the automaticity may be due to the maintenance state of WM, and therefore reflects an effect of active executive processing.

To decode the two states of WM from attention-related ERP response, the SVM classifier was trained and resulted in an accuracy of 68.18%. An MVPA approach can extract an identifying pattern of brain activity by constructing a classifier that maximally discriminates between mental states [18, 38, 39]. We applied multivariate patterns of ERP components (P1 and N2pc) to predict the WM states in modulating visual attention, as those components were commonly used to index selective attention [8, 14]. Thus, both P1 and N2pc components are important for identifying the WM states, which may help to build a reliable objective neuronal marker to explore the interaction between the WM states (maintenance and inhibition) and visual attention.

In addition, although the proportion between the presence and absence of the cued color was same in the memory test, participants showed better performance in the prior condition. Remembered contents would be stored in WM representation [5, 32], especially when those cued contents were related to the subsequent task. The representation of the cued items might be in an active state [3]. As indicated by previous study, visual WM capacity is a limited resource that can be dynamic shifted between objects [40]. The precision of WM is determined by the proportion of resource allocation to each object. In the present study, to-be-maintained items would have higher precision of WM than to-be-inhibited items. Therefore, whether the cued items appeared would induce unbalanced performance in the memory test. Besides, we also found that the uncued items might be mistaken for the cued items when the cued items were absent, suggesting that both the cued and uncued items existed in WM. However, they had distinct influence on visual search, which might be caused by the different states of remembered items retaining in WM.

These findings suggest a possible reinterpretation of the influence of WM on selective attention that is often described as top-down modulation [12]. We propose that the two aspects of WM representation should play different roles in the guidance effect: remembered content may be an anchor for the shift of attention, while the state of memorized items retaining in WM may maintain access from WM content to input information. The current findings highlight the roles of inhibition and maintenance in the guidance of WM on visual attention, in which to-be-maintained items function as an actively attentional template which may directly capture the attention and further impair the search for target. By contrast, to-be-inhibited items may be viewed as a "negative template" and can facilitate the search for target. Given that the other negative template studies involved an actively-maintained negative template that is being actively utilized during search [28, 41], future research should explore the role of WM inhibition in the active condition. Finally, as compared to univariate methods, the multivariate approach built a reliable instrument to explore the complex patterns of working memory in modulating visual attention.

In summary, by assembling to-be-maintained and to-be-inhibited colors in a common WM template, and especially utilizing the opposite influences of the inhibition and remembered content in WM on visual attention, we investigated the roles of the WM states in the guidance of WM during visual search. As the color matching the remembered content in WM

which was always located on the side contralateral to the target, to-be-maintained colors impaired the visual search by guiding attention to the contralateral side but not diminishing early sensory processing. In contrast, to-be-inhibited colors facilitated visual search by diminishing sensory processing but not guiding attention to the contralateral side.

## Author Contributions

**Conceptualization:** Antao Chen.

**Data curation:** Ting Luo, Na Hu.

**Formal analysis:** Quanshan Long, Yuanling Jiang, Peng Xu.

**Methodology:** Ting Luo, Sheng Zhang.

**Supervision:** Antao Chen.

**Visualization:** Quanshan Long.

**Writing – original draft:** Quanshan Long.

**Writing – review & editing:** Quanshan Long, Yan Gu, Antao Chen.

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
