## [Decision Letter · Decision Letter 0]

26 Aug 2020

PONE-D-20-19909

The Roles of Executive Operations in the Guidance of Working Memory on Visual Attention

PLOS ONE

Dear Dr. Chen,

Thank you for submitting your manuscript to PLOS ONE. After careful consideration, we feel that it has merit but does not fully meet PLOS ONE’s publication criteria as it currently stands. Therefore, we invite you to submit a revised version of the manuscript that addresses the points raised during the review process.

Both reviewers agree that the study were well conducted in general. On the other hand, there are some places to be refined in terms of experimental settings and result explanation. I think these comments are direct and I will not repeat them here. 

We look forward to receiving your revised manuscript.

Kind regards,

Zaifeng Gao

Academic Editor

PLOS ONE

Journal Requirements:

Reviewers' comments:

Reviewer's Responses to Questions

**Comments to the Author**

1. Is the manuscript technically sound, and do the data support the conclusions?

Reviewer #1: Yes

Reviewer #2: Yes

2. Has the statistical analysis been performed appropriately and rigorously? 

Reviewer #1: No

Reviewer #2: Yes

3. Have the authors made all data underlying the findings in their manuscript fully available?

Reviewer #1: Yes

Reviewer #2: No

4. Is the manuscript presented in an intelligible fashion and written in standard English?

Reviewer #1: Yes

Reviewer #2: No

5. Review Comments to the Author

Reviewer #1: Title: The Roles of Executive Operations in the Guidance of Working Memory on Visual Attention

The current study explored whether state of memorized items retaining in working memory (WM) influence attentional guidance when these items are presented as distractors in visual search task. To do so, the author employed a WM change-detection task and inserted a visual search task in maintenance phase, with EEG records. Participants were first required to memorize two colors and then maintenance the cued one as well as inhibit the uncued one. Then they were required to do a visual search task in which either the cued or uncued color could be distractors. Results showed that in visual search task, uncued condition had shorter search time and attenuated P1 amplitude than neutral condition, while cued condition had longer search time and enhanced N2pc amplitude than neutral condition. Furthermore, multivariate pattern analyses could classify cued and uncued conditions accurately. Together the authors suggest that the active item in WM could guide attention by accessing input information, while the inhibit item in WM could deter the shifting of attention by suppressing sensory processing.

Overall, the current study addressed an interesting question, the whole manuscript (ms) was elegantly organized, the data is solid. However, I have a few concerns about the current ms which discourage my enthusiasm. I will list them below regardless of importance.

Major Points:

1) The ms described current question as “whether the operations of WM contribute to attentional guidance when the contents remembered in WM are presented as distractor in the search task”. In terms of experimental procedure, I think the more accurate describe of current question is about state of memorized items retaining in WM instead of operations of WM. On the other hand, the current study focuses on the influence of distractors in a visual search task when the distractors are items retaining in a related WM task. So I would suggest the author added the information of distractors in title to make it more focal.

2) In line 200, “for multiple comparisons, p values were adjusted using the Least Significant Difference (LSD) correction.” The more appropriate way for multiple comparisons is Bonferroni correction.

3) In line 230, “a classifier was trained on 43 datasets and 231 then applied to the remaining dataset”. What the 44 datasets were? In context, the 44 datasets seems from 22 participants × 2 conditions (maintenance vs. inhibition). If so, I would suggest the author combining data from the same participant and treat it as one dataset, then train on 21 datasets from 21 participants and test the remaining dataset using the leave one-out cross-validation method. Because data from the same participants may have common characteristics related to the participants, which may lead to unexpected feature extraction in SVM.

4) WM task in ms is to judge whether the cued item is presented in memory test. In this task, proportion of presence and proportion of absence are the same, but task performance in cued presence condition and cued absence condition are quite different. I would suggest the author to discuss this unbalanced performance.

Minor Points:

5) In line 158, dose “hemisphere” means hemisphere of brain? Or does it means left side or right side of the screen?

6) Behavioral results in ms only report interaction now. I would suggest the author to report intact ANOVA results with main effects as well as interaction.

7) It will be a better exhibition of data in Figure 4 if add a brain electrical activity mapping like in Figure 3.

Reviewer #2: The present study presents an intriguing finding. I don't find any serious flaw in experimental design and data analyses.

However, I have several concerns regarding the interpretation of results and discussion.

First, not only the uncued item, but the cued item should also play a role in formulating the template for rejection. In the experiment, working memory items, maintained or inhibited, were always presented contralaterally to the target. Hence, any memory-matching stimulus is a distractor. In this case, the cued item should also be regarded as a template for rejection in visual search. Certainly, the cued item should be maintained in working memory, evoking a conflict between search demand and working memory task. This conflict might have hindered the effect of rejection template. For the uncued item, due the inhibitory tagging imposed on the stimulus, the rejection template could be established more effectively.

Second, it is nice to see that the amplitudes of the Pd and N2pc well match behavioral results. Contrary to this, I don't see any merit in the MVPA analyses. What does it add to the behavioral results and univariate ERP results? The cued item should be maintained in working memory and the uncured item is not in working memory. It is not surprising that this difference would yield huge difference in behavior and neural responses.

As a minor issue, English in the manuscript is not too bad, but it has to be more scholarly. Some of expressions are too colloquial and not completed sentences.

6. PLOS authors have the option to publish the peer review history of their article (what does this mean?). If published, this will include your full peer review and any attached files.

Reviewer #1: No

Reviewer #2: No

---

## [Author Response · Author response to Decision Letter 0]

8 Oct 2020

Authors’ Response to the Review Comments

Response to Comments from Reviewer 1

Reviewer #1: Title: The Roles of Executive Operations in the Guidance of Working Memory on Visual Attention

The current study explored whether state of memorized items retaining in working memory (WM) influence attentional guidance when these items are presented as distractors in visual search task. To do so, the author employed a WM change-detection task and inserted a visual search task in maintenance phase, with EEG records. Participants were first required to memorize two colors and then maintenance the cued one as well as inhibit the uncued one. Then they were required to do a visual search task in which either the cued or uncued color could be distractors. Results showed that in visual search task, uncued condition had shorter search time and attenuated P1 amplitude than neutral condition, while cued condition had longer search time and enhanced N2pc amplitude than neutral condition. Furthermore, multivariate pattern analyses could classify cued and uncued conditions accurately. Together the authors suggest that the active item in WM could guide attention by accessing input information, while the inhibit item in WM could deter the shifting of attention by suppressing sensory processing.

Overall, the current study addressed an interesting question, the whole manuscript (ms) was elegantly organized, the data is solid. However, I have a few concerns about the current ms which discourage my enthusiasm. I will list them below regardless of importance.

Major Points:

1) The ms described current question as “whether the operations of WM contribute to attentional guidance when the contents remembered in WM are presented as distractor in the search task”. In terms of experimental procedure, I think the more accurate describe of current question is about state of memorized items retaining in WM instead of operations of WM. On the other hand, the current study focuses on the influence of distractors in a visual search task when the distractors are items retaining in a related WM task. So I would suggest the author added the information of distractors in title to make it more focal.

Response: We greatly appreciated the reminder about this important issue. We totally agree that using “the state of rememorized items retaining in working memory” to describe the current question is more accurate. According to your advice, we revised the title of the manuscript and relevant content in the manuscript.

The new title is that “The state of memory-matched distractor in working memory influence the visual attention”.

“An intriguing question is whether the state of memorized items retaining in WM (e.g., inhibition and maintenance) contribute to this attentional guidance when the memory-matched items are presented as distractor in the search task.” (Introduction. Line 56-58)

“In the present study, we planned to use a revised dual task to investigate the influences of the state (especially, inhibition) of memorized items retaining in WM on visual attention.” (Introduction. Line 80-82)

2) In line 200, “for multiple comparisons, p values were adjusted using the Least Significant Difference (LSD) correction.” The more appropriate way for multiple comparisons is Bonferroni correction.

Response: Thanks for your reminder. We conducted the new data analyses, in which we adjusted p values by using Holm-Bonferroni correction instead of using LSD in the revised manuscript. And the new results of behavior and ERP were similar to previous results. The detailed results can be found in Results.

According to your advice, we revised the manuscript as below:

“For multiple comparisons, the Holm-Bonferroni correction was used to adjusted p value.” (Data analyses. Line 205-206)

3) In line 230, “a classifier was trained on 43 datasets and 231 then applied to the remaining dataset”. What the 44 datasets were? In context, the 44 datasets seems from 22 participants × 2 conditions (maintenance vs. inhibition). If so, I would suggest the author combining data from the same participant and treat it as one dataset, then train on 21 datasets from 21 participants and test the remaining dataset using the leave one-out cross-validation method. Because data from the same participants may have common characteristics related to the participants, which may lead to unexpected feature extraction in SVM.

Response: Thank you very much. In order to avoid unexpected feature extraction in SVM, we used a cross-subject approach to maximally discriminate between inhibition and maintenance in the new data analyses. To this end, we combined data from the same participant in the new analyses. A classifier was trained on the data of 21 subjects and then applied to the remaining subject by using the leave-one-out cross-validation method. The accuracy of classification was 68.18% (p = 0.028), suggesting that inhibition and maintenance are significantly different with each other in modulating visual attention.

According to your suggestion, we revised the corresponding contents as below:

“The MVPA was performed to maximally discriminate between inhibition and maintenance. A cross-subject approach was used in the present study, in which a classifier was trained on the data of 21 subjects and then applied to the remaining subject by using the leave-one-out cross-validation method. We applied an MVPA combining feature selection techniques with a support-vector machine (SVM) by using the LIBSVM toolbox [26].” (Methods. Data analyses. Line 233-238)

“First, the SVM classifier was trained on both ERP difference waves, resulting in a maximum accuracy of 68.18%. Permutation tests confirmed that this accuracy was significantly greater than chance (50%; p = 0.028).” (Results. MVPA analyses. Line 338-340)

4) WM task in ms is to judge whether the cued item is presented in memory test. In this task, proportion of presence and proportion of absence are the same, but task performance in cued presence condition and cued absence condition are quite different. I would suggest the author to discuss this unbalanced performance.

Response: Thank you for the insightful suggestion. We think that the unbalanced performance between the cued and the uncued conditions might be caused by different precision of working memory in the memory test. The visual working memory capacity is limited, and the proportion of resource allocation to each object determines the precision of working memory (Bays & Husain, 2008). In the revised manuscript, we discussed this important point.

According to your advice, we added the content to the Discussion.

“In addition, although the proportion between the presence and absence of the cued color was same in the memory test, participants showed better performance in the prior condition. Remembered contents would be stored in WM representation [5,32], especially when those cued contents were related to the subsequent task. The representation of the cued items might be in an active state [3]. As indicated by previous study, visual WM capacity is a limited resource that can be dynamic shifted between objects [40]. The precision of WM is determined by the proportion of resource allocation to each object. In the present study, to-be-maintained items would have higher precision of WM than to-be-inhibited items. Therefore, whether the cued items appeared would induce unbalanced performance in the memory test. Besides, we also found that the uncued items might be mistaken for the cued items when the cued items were absent, suggesting that both the cued and uncued items existed in WM. However, they had distinct influence on visual search, which might be caused by the different states of remembered items retaining in WM.” (Discussion. Line 443-456)

Reference:

3. Olivers CN, Peters J, Houtkamp R, Roelfsema PR (2011) Different states in visual working memory: when it guides attention and when it does not. Trends in Cognitive Sciences 15: 327-334.

5. Olivers CN, Meijer F, Theeuwes J (2006) Feature-based memory-driven attentional capture: visual working memory content affects visual attention. Journal of Experimental Psychology: Human Perception and Performance 32: 1243.

32. Kornblith S, Quiroga RQ, Koch C, Fried I, Mormann F (2017) Persistent single-neuron activity during working memory in the human medial temporal lobe. Current Biology 27: 1026-1032.

40. Bays PM, Husain M (2008) Dynamic shifts of limited working memory resources in human vision. Science 321: 851-854.

Minor Points:

5) In line 158, dose “hemisphere” means hemisphere of brain? Or does it means left side or right side of the screen?

Response: Thank you for reminding us about this important point. Participants were introduced to discriminate the locations of targets. If the targets were presented on the left or right side of the screen, they pressed the “F” or “J” key. Below is the revised content:

“Participants discriminated the location of targets where the tilted line appeared by pressing keyboard (“F” for left side of the screen, “J” for right side of the screen).” (Apparatus, stimuli, and procedure. Line 162-164)

6) Behavioral results in ms only report interaction now. I would suggest the author to report intact ANOVA results with main effects as well as interaction.

Response: Thanks for your suggestion. In the revised manuscript, we reported the main effect and interaction in the behavioral results. 

According to your advice, we reported the main effect in the analyses of the accuracy as below:

“Also, the main effect of Presence of cued was significant (F(1,21) = 24.71, p < 0.001, ηp2 = 0.54), with higher accuracy in the presence (91.5%) than in the absence of the cued color (77.4%). And the main effect of Presence of uncued were significant (F(1,21) = 29.64, p < 0.001, ηp2 = 0.58), with lower accuracy in the presence (82.0%) than in the absence (86.9%) of the uncued color.” (Results. Behavioral results. Memory test. Line 270-275)

Besides, we also reported the main effect in the analyses of the response time as below:

“The main effect of Presence of cued was significant (F(1,21) = 10.57, p = 0.004, ηp2 = 0.34), with faster response in the presence (807 ms) than in the absence (876 ms) of the cued color. The main effect of Presence of uncued was not significant (F(1,21) = 0.05, p = 0.831).”(Results. Behavioral results. Memory test. Line 284-287)

7) It will be a better exhibition of data in Figure 4 if add a brain electrical activity mapping like in Figure 3.

Response: Thanks for your advice. According to your suggestion, we added the Topography of the N2pc component in Fig.4 as below:

Fig. 4. The N2pc waveforms (a, b, c) for contralateral and ipsilateral electrical activity and mean difference wave (d) are presented. Topography represents the N2pc component (170-250 ms). The left and right sides of the outline head represent ipsilateral and contralateral electrode sites of the target, respectively. *** p < .001. (Line 332-335)

Response to Comments from Reviewer 2

Reviewer #2: The present study presents an intriguing finding. I don't find any serious flaw in experimental design and data analyses.

However, I have several concerns regarding the interpretation of results and discussion.

First, not only the uncued item, but the cued item should also play a role in formulating the template for rejection. In the experiment, working memory items, maintained or inhibited, were always presented contralaterally to the target. Hence, any memory-matching stimulus is a distractor. In this case, the cued item should also be regarded as a template for rejection in visual search. Certainly, the cued item should be maintained in working memory, evoking a conflict between search demand and working memory task. This conflict might have hindered the effect of rejection template. For the uncued item, due the inhibitory tagging imposed on the stimulus, the rejection template could be established more effectively.

Response: Thank you for the insightful comment. It is possible that the memory-matched colors (cued and uncued colors) were regarded as a template for rejection in the search task. The search demand and the requirement of working memory might engender the same response in the cued condition; but they might engender the different response in the uncued condition, which would evoke conflict (Kerns et al., 2004; Macleod, 1991; Weissman et al., 2017). Thus, the time of visual search significantly increased in the cued condition but significantly speeded up in the uncued condition, compared to the neutral condition.

According to your advice, we added the content to the Discussion.

“In accordance with previous studies [1,4,5,8], our behavioral results showed that the to-be-maintained color in the cued condition impaired the visual search. In the present study, the memory-matched color was a distractor that was always presented contralateral to the search target. The to-be-maintained items might be viewed as a template for rejection in the search task [10], while these items should be maintained in the working memory task. In this case, the requirement of working memory and search demand engender different response, evoking a conflict [34-36]. Therefore, this conflict hindered the effect of rejection template, leading to slower response time in the cued condition than in the neutral condition.” (Discussion. Line 408-416)

Reference

10. Reeder RR, Olivers CN, Pollmann S (2017) Cortical evidence for negative search templates. Visual Cognition 25: 278-290.

34. Macleod CM (1991) Half a Century of Research on the Stroop Effect: An Integrative Review. Psychological Bulletin 109: 163-203.

35. Weissman DH, Colter KM, Grant LD, Bissett PG (2017) Identifying Stimuli That Cue Multiple Responses Triggers the Congruency Sequence Effect Independent of Response Conflict. Journal of Experimental Psychology-Human Perception and Performance 43: 677-689.

36. Kerns JG, Cohen JD, MacDonald AW, Cho RY, Stenger VA, et al. (2004) Anterior Cingulate conflict monitoring and adjustments in control. Science 303: 1023-1026.

Second, it is nice to see that the amplitudes of the Pd and N2pc well match behavioral results. Contrary to this, I don't see any merit in the MVPA analyses. What does it add to the behavioral results and univariate ERP results? The cued item should be maintained in working memory and the uncured item is not in working memory. It is not surprising that this difference would yield huge difference in behavior and neural responses.

Response: Thank you for comments on this important issue. In the current study, we want to explore whether the state of memorized items contribute to attentional guidance when the contents remembered in WM are presented as distractor in the search task. To this end, we conducted two different methods of data analyses. On the one hand, behavioral and univariate results could describe the difference between cued and uncued items in the level of single component. On the other hand, we combined all ERP components to investigate their difference from the multivariate level by using the MVPA analyses. MVPA analysis may combine all the characteristics of the data to help us understand the differences from a holistic perspective.

Importantly, the results of MVPA showed that the accuracy is maximum when the SVM classifier was trained on both ERP (P1 and N2pc). These findings suggested that the difference of neural responses between maintenance and inhibition dependent on the multivariate characteristics. Overall, these results might provide more evidence to understand the different state of memorized items retaining in working memory.

In order to clarify the role of MVPA analyses, we added some contents in the Introduction about this point.

“Given that multivariate pattern analyses (MVPA) [18] can combine all characteristics of the data to help us understand the difference between inhibition and maintenance from the multivariate level, we used MVPA to maximally discriminate two WM states in modulating attention.” (Introduction. Line 94-97)

As a minor issue, English in the manuscript is not too bad, but it has to be more scholarly. Some of expressions are too colloquial and not completed sentences.

Response: Thanks for your careful check. We feel sorry for the inappropriate description. We have improved the clarity and readability of the manuscript.

---

## [Decision Letter · Decision Letter 1]

9 Nov 2020

The state of memory-matched distractor in working memory influence the visual attention

PONE-D-20-19909R1

Dear Dr. Chen,

We’re pleased to inform you that your manuscript has been judged scientifically suitable for publication and will be formally accepted for publication once it meets all outstanding technical requirements.

Kind regards,

Zaifeng Gao

Academic Editor

PLOS ONE

Additional Editor Comments (optional):

Reviewers' comments:

Reviewer's Responses to Questions

**Comments to the Author**

1. If the authors have adequately addressed your comments raised in a previous round of review and you feel that this manuscript is now acceptable for publication, you may indicate that here to bypass the “Comments to the Author” section, enter your conflict of interest statement in the “Confidential to Editor” section, and submit your "Accept" recommendation.

Reviewer #1: All comments have been addressed

Reviewer #2: All comments have been addressed

2. Is the manuscript technically sound, and do the data support the conclusions?

Reviewer #1: (No Response)

Reviewer #2: Yes

3. Has the statistical analysis been performed appropriately and rigorously? 

Reviewer #1: (No Response)

Reviewer #2: Yes

4. Have the authors made all data underlying the findings in their manuscript fully available?

Reviewer #1: (No Response)

Reviewer #2: Yes

5. Is the manuscript presented in an intelligible fashion and written in standard English?

Reviewer #1: (No Response)

Reviewer #2: Yes

6. Review Comments to the Author

Reviewer #1: I think the authors have adequately addressed the comments raised in a previous round of review. I don't find any serious flaw in the manuscript.

Reviewer #2: I think that the authors well addressed all of my issues. I think that the paper is now suitable for publication. I have no further comment.

7. PLOS authors have the option to publish the peer review history of their article (what does this mean?). If published, this will include your full peer review and any attached files.

Reviewer #1: **Yes: **Xiqian Lu

Reviewer #2: **Yes: **suk won han

---

## [Editor Report · Acceptance letter]

17 Nov 2020

PONE-D-20-19909R1 

The state of memory-matched distractor in working memory influence the visual attention 

Dear Dr. Chen:

I'm pleased to inform you that your manuscript has been deemed suitable for publication in PLOS ONE. Congratulations! Your manuscript is now with our production department. 

Kind regards, 

on behalf of

Professor Zaifeng Gao 

Academic Editor

PLOS ONE